# Design and Synthesis of π-Extended Resveratrol Analogues and In Vitro Antioxidant and Anti-Inflammatory Activity Evaluation

**DOI:** 10.3390/molecules26030646

**Published:** 2021-01-26

**Authors:** Ki Yoon Nam, Kongara Damodar, Yeontaek Lee, Lee Seul Park, Ji Geun Gim, Jae Phil Park, Seong Ho Jeon, Jeong Tae Lee

**Affiliations:** 1Department of Chemistry and Institute of Applied Chemistry, Hallym University, Chuncheon 24252, Korea; gugu4478@naver.com (K.Y.N.); kongaradamu@gmail.com (K.D.); hellobong3@nate.com (L.S.P.); M17515@hallym.ac.kr (J.G.G.); 2Department of Life Science and Multidisciplinary Genome Institute, Hallym University, Chuncheon 24252, Korea; M18026@hallym.ac.kr (Y.L.); pjp8683@hallym.ac.kr (J.P.P.)

**Keywords:** resveratrol, extended aromaticity, antioxidant, anti-inflammatory, cell viability

## Abstract

The research on resveratrol (**1**) has been conducted intensively over a long time due to its proven antioxidant activity and disease-fighting capabilities. Many efforts have also been made to increase these biological effects. In the present study, six new extended aromatic resveratrol analogues containing naphthalene (**2**) and its bioisosteres quinoline (**3** and **4**), isoquinoline (**5**) quinoxaline (**6**) and quinazoline (**7**) scaffolds were designed and synthesized using an annulation strategy. The antioxidant and anti-inflammatory activities of these compounds were investigated. All compounds showed better antioxidant activity than resveratrol in ABTS assay. As for the anti-inflammatory test, **5** and **7** exhibited better activity than resveratrol. It is worth noting that nitrogen substitution on the extended aromatic resveratrol analogues has a significant impact on cell viability. Taking the antioxidant activities and NO inhibition activities into consideration, we conclude that isoquinoline analogue **5** may qualify for the further investigation of antioxidant and anti-inflammatory therapy. Furthermore, our study results suggest that in order to improve the biological activity of polyphenolic compounds, extended aromaticity and nitrogen substitution strategy could be a viable method for the design of future drug candidates.

## 1. Introduction

The oxygen element is essential for human life, but under certain circumstances, it also has harmful effects. Free radicals are known as reactive oxygen species (ROS), generated by the oxygen metabolism in the human body, and have many biochemical activities such as cell growth, proliferation, and survival at their physiological concentrations [1]. Nevertheless, excessive production of ROS and reactive nitrogen species (RNS) causes damage to DNA, cellular proteins, carbohydrates, and membrane lipids and eventually cell injury and death. The human body’s protective attempt against this damage is achieved by endogenous antioxidant enzymes (superoxide dismutase, catalase, and glutathione peroxidase) or nonenzymatic antioxidants (uric acid, bilirubin, lipoic acid, metallothioneins) [2]. The antioxidant component can interact with ROS/RNS to stop the radical chain reactions, and thereby delay or block cellular damage. When the balance between free radical generation and antioxidant defenses is disturbed, the situation is called oxidative stress (OS) [3]. OS causes various pathological conditions such as cancer, hypertension, neurological disorder, and diabetes. When the endogenous antioxidant system fails to protect the organism against ROS/RNS damage, exogenous antioxidants are needed as diet, nutritional supplements or pharmaceutical products, which contain antioxidants as the principle active compound [4].

Inflammation is a natural response of living tissue against injury or infection. Activated inflammatory pathway generates several inflammatory mediators such as cytokines (IL-1β, IL6 and TNF-α, chemokines, free radicals, NO (nitric oxide) and eicosanoids (prostaglandins)) in the affected tissues that eventually guide the restoration of the normal tissue structure. However, their secretion for an extended period can lead to OS and chronic inflammation-associated disorders. Excess ROS/RNS can trigger an intracellular signaling cascade that augments pro-inflammatory gene expression [5]. Moreover, the upsurge of NO magnitude by macrophages during the inflammatory response leads to undesirable tissue destruction [6]. Hence, inflammation and OS are closely related to pathophysiological events [7]. This mutual interdependence has a crucial role in the pathogenesis and management of various diseases, and we may expect beneficial effects from the antioxidant and anti-inflammatory therapy.

Resveratrol (3,5,4′-trihydroxy-*trans*-stilbene) (**1**) (Figure 1) is a natural polyphenolic stilbenoid produced in some plants as a phytoalexin. It is widely recognized as a nutraceutical and therapeutic compound given its positive effects on human health. Promising activities of **1** include antioxidant, anti-aging, anticancer, anti-inflammatory, antidiabetic, and cardioprotective properties [8,9,10]. Antioxidant activity of **1** is due to its substantial scavenging property of ROS, RNS and other secondary organic radicals formed by the reaction of ROS and RNS with biomolecules. It also acts as a useful metal ions chelator, maintains the oxidation-reduction balance in a cell, and reduces the activity of the enzymes which involve in ROS production. However, its potential application in some cancer studies was ambiguous due to its poor bioavailability [11].

Previous studies of **1** have evidenced that the presence of three hydroxy groups in 3, 5 and 4′ positions, presence of aromatic rings and a trans-double bond between the aromatic rings, are vital for its antioxidant activity [12,13]. Analogues of **1** bearing electron-donating groups on the benzene rings displayed increased its antioxidant activity [14]. The free radical scavenging property undoubtedly depends on the resonance stabilization of the resulting phenoxy radicals. Hence, an extension of π-conjugation could lead to the better stabilization of the phenoxy radical, which can result in lowering the phenolic O–H bond dissociation energy, thereby increased antioxidant activity can be obtained [15]. Zhou et al. applied this concept in designing analogues of **1** by introducing additional double bonds between two benzene rings and observed increased antioxidant activity compared to **1 [16]**. Another promising approach to achieve the extended π-conjugation would be extended aromaticity strategy. It is also apparent that π-delocalization is maximized in cyclic π-systems compared to acyclic ones because the former systems contain extended π-surfaces [17]. We believe that no report has been observed in the literature based on the π-system elongation of **1** through aromaticity extension in a systematic way. 

According to a recent report, heterocyclic replacements for benzene can have a significant impact on ADME (absorption, distribution, metabolism, and excretion) -related parameters [18]. Many biologically active natural products such as alkaloids, niacin, and vitamin B_6_ contain nitrogen. These two facts inspired us to investigate the effect of nitrogen incorporation in extended aromatic compounds on biological activity. In this context, we designed and synthesized new analogues of **1** with extended aromaticity in which the 4’-hydroxy group-containing benzene ring is fused with benzene (naphthalene analogue **2**) and its bioisosteres pyridine (quinoline and isoquinoline analogues **3**, **4** and **5**), pyrazine (quinoxaline analogue **6**) and pyrimidine (quinazoline analogue **7**) scaffolds (Figure 1), and investigated their trends in their antioxidant and anti-inflammatory activity with respect to the extended aromaticity as well as the substitution with nitrogen atoms.

## 2. Results and Discussion

### 2.1. Chemistry

Synthesis of resveratrol analogues **2**–**7** commenced with the protection of phenolic hydroxyl groups of 3,5-dihydroxybenzaldehyde **8** (Scheme 1). Treatment of **8** with benzyl bromide (BnBr) in the presence of K_2_CO_3 _ afforded bis(Bn) ether **9** which was further transformed to the corresponding phosphonate ester **10** after reduction, bromination and Arbuzov reactions on the aldehyde group of **9**, successively [19]. Naphthalene aldehyde **13**, which was used for the resveratrol analogue synthesis, was obtained from 6-hydroxy-2-naphthoic acid **11**. Treatment of **11** with BnBr/K_2_CO_3 _ resulted in compound **12**, which was subjected to reduction and oxidation synthetic sequence to provide the aldehyde **13** in high yields. For the synthesis of quinoline-3-carbaldehyde moiety **16**, we employed Vilsmeier–Haack reaction. 3-Benzyloxyacetanilide **14** efficiently underwent Vilsmeier–Haack cyclization [20] to afford 2-chloro-3-formyl quinoline derivative **15** and subsequent hydrodechlorination of **15** was realized to yield **16** using Pd(PPh_3_)_4_/Et_3_N/HCO_2 _H system in DMF at 110 °C. Next, a metal-free, mild and efficient method was employed for the synthesis of 2-methylquinoline moiety **18** by condensation of 4-(benzyloxy)aniline **17** with ethyl vinyl ether in the presence of catalytic amount of iodine [21]. To achieve isoquinoline moiety **23**, Heck reaction was performed on 5-(benzyloxy)-2-bromobenzaldehyde **20** which in turn obtained from 2-bromo-5-hydroxybenzaldehyde **19**. Treatment of **20** with methyl ester of *N*-acetyl dehydroalanine using Pd(OAc)_2_/P(*o*-tol)_3_/TEA system in DMSO underwent Heck reaction, whereupon a spontaneous intramolecular transesterification resulted in the isoquinoline-3-carboxylate derivative **21**. Subsequent reduction to the corresponding alcohol **22** and oxidation with IBX gave the isoquinoline aldehyde **23**. 1,4-Difluoro-2-nitrobenzene **24** was used as a precursor to obtain the quinoxaline moiety **29 [22]**. Nucleophilic displacement reaction of **24** with l-alanine ethyl ester hydrochloride furnished **25**, which subsequently underwent cyclization to give quinoxalone **26**. When **26** was subjected to Vilsmeier–Haack condition, quinoxaline moiety **27** was formed which finally transformed to the desired moiety **29** by hydrodechlorination and nucleophilic displacement reaction synthetic sequences, respectively. The requisite quinazoline moiety **30** was constructed by copper-catalyzed cascade coupling of **20** with acetamidine HCl [23].

With all the key fragments secured, their linkage to construct analogues of **1** was then executed (Scheme 2). At first, we tried the Wittig–Horner reaction between **13** and **10** using NaH in DMF, and the resulting benzyl protected naphthalene based **1** analogue **31** was obtained in high yield. In a similar manner, quinoline and isoquinoline analogues **32** and **33** were also acquired. Next, compounds **34**–**36** were built up through C(sp^3^)–H functionalization of methyl azaarenes **18**, **29** and **30** with the aldehyde **9**, respectively [24]. Notably, excellent stereoselectivity (based on ^1^H-NMR) was achieved in both conditions. Finally, benzyl (Bn) group deprotection of **31**–**35** using 1.0 M BBr_3_ (in CH_2_Cl_2 _) smoothly furnished the desired target compounds **2**–**6**, respectively. Unfortunately, very poor yield was observed for quinazoline-based analogue **7** even after extended reaction time. To solve this problem, we employed AlCl_3_/*N*,*N*-dimethylaniline system [25]. This reagent system afforded the product **7** in 3 h, albeit in low yield. All the structures of the final products **2**–**7** were settled from their spectral data (^1^H and ^13^C NMR and MS) (see the Appendix A).

### 2.2. Antioxidant Activity

ABTS (2,2′-azinobis (3-ethylbenzothiazoline-6-sulfonic acid) diammonium salt) assay was used for the antioxidant estimation. Though many methods are available, this appears to be the most popular because of its operational simplicity, short experimental time, and the employment of the inexpensive spectrophotometer [26]. In vitro ABTS radical scavenging activities of **1** and its analogues **2**–**7** are summarized in Table 1. The table illustrates that all the analogues **2**–**7** showed excellent ABTS free radical scavenging activity with better IC_50_ values (43~23% better activity, 3.14 ± 0.445, 3.57 ± 0.496, 3.61 ± 0.253, 2.81 ± 0.034, 3.10 ± 0.953 and 3.54 ± 0.714 μM, respectively) than the standard/positive control **1** (4.88 ± 0.35 μM). We believe that the high ABTS radical scavenging activity of **2**–**7** was due to the enhanced resonance stabilization of the resulting phenoxy radicals. Compared to **1**, all the analogues **2**–**7** showed bicyclic aromatic moieties instead of a benzene ring, and this feature facilitated the extended and planar π conjugation by which the stronger capability to stabilize the radical species via delocalization of the π-electrons can be achieved. The lower the IC_50_ value of a compound, the higher its antioxidant activity. Compound **5**, which contains an isoquinoline moiety, expressed the best activity among all analogues. The nitrogen in the Y position (such as compounds **5** and **7** in Scheme 2) increased the antioxidant activity, and the nitrogen in the Z position (such as compounds **4** and **6** in Scheme 2) decreased the antioxidant activity. However, there appears to be no clear correlation between antioxidant activity and the number of nitrogen atom substitutions in the ABTS assay results. It seems that the position of nitrogen substitution is more important than the number of nitrogen substitution on the activity. 

### 2.3. Anti-Iinflammatory Effects on Macrophage

In the pathogenesis of inflammatory-related disorders, NO is recognized as one of the important mediators. Hence, its inhibition is considered as a beneficial approach for the prevention of inflammation [27]. Herein, to understand the anti-inflammatory properties of the resveratrol analogues, we studied their inhibitory effects against NO production in LPS-treated RAW 264.7 cells. Although it is not possible to measure NO quantitatively in a biological environment due to its rapid oxidation to NO_2_ (nitrite), short physiological half-life and presence of other free radicals, NO secretion activity in cells is indirectly determined by Griess assay counting its stable product NO_2_ levels [28]. The inhibitory effects of **1**–**7** on NO production were collected in Table 2. Pretreatment with **1** and its analogues **2**–**7** showed some reduction in the NO production dose-dependently in LPS-treated cells (see Appendix A). The median effective doses (ED_50_) for inhibiting the production of NO by **1**–**7** were calculated on the basis of NO_2_ released into the culture media and were observed in the range of 13.08–56.88 μM. The activities of **5** (ED_50_ = 13.08 μM) and **7** (ED_50_ = 21.67 μM) were superior to that of **1** (26.89 ± 1.210), while the activities of **2** (ED_50_ = 29.50 μM), **3** (ED_50_ = 44.60 μM), **4** (ED_50_ = 38.81 μM) and **6** (ED_50_ = 56.88 μM) were inferior to that of **1**. Notably, isoquinoline analogue **5** exerted the highest reduction in NO production. Next, an assessment of cellular metabolic activity was conducted by MTT assay to prove that inhibitory effects of **1**–**7** on LPS-stimulated NO secretion was not associated with their cytotoxic effect. Cell viability was not affected up to 50 μM except for **2**. We also applied in vitro efficacy index (*i*EI) paradigm [29], the ratio of toxicity (LD_50_)/anti-inflammatory potency (ED_50_), to confirm that the anti-inflammatory activity is in the viable cells only. All nitrogen substituted extended resveratrol analogues **3**–**7** (*i*EI = 2.11~4.84) are more potent than their non-nitrogenated bioisostere **2** (*i*EI = 0.98); especially, compounds **3**–**5** and **7** showed better *i*EI values than even the parent compound **1**. This result leads us to conclude that nitrogen substitution of extended aromatic resveratrol conferred a significant advantage over non-substituted extended aromatic resveratrol in terms of cytotoxicity. On the whole, compounds **5** and **7** have much-improved *i*EI values compared to any other compounds. When we combine the results from the antioxidant activity and anti-inflammatory activity, we may conclude that isoquinoline analogue **5** had a strong correlation between antioxidant and anti-inflammatory effects (Table 1 and Table 2) and could be the lead compound for both antioxidant and anti-inflammatory agents.

We have further investigated whether resveratrol and its analogues can reduce the inflammatory activity of macrophages. For this purpose, we examined the downregulation of pro-inflammatory cytokine gene expression of LPS-stimulated RAW 264.7 cells by potent inhibitors of NO production (**1**, **2**, **5**, and **7**) (Figure 2). As expected, the levels of IL-1β and IL-6 were significantly reduced by treatment of resveratrol and its analogues. On the other hand, the level of TNF-α did not change upon treatment. Treatment with 50 μM of analogue **2** was highly toxic to the cells. Since most of the RNA was degraded by cell death, gene amplification by PCR did not occur.

## 3. Materials and Methods

All starting materials, chemical and biological reagents, and solvents were purchased from commercial sources and used without any additional purification. Solvents used for reactions were purchased as either anhydrous grade products or freshly distilled using proper dehydrating agents. ^1^H NMR and ^13^C NMR spectra were recorded at room temperature on a Varian Mercury TM 300 MHz FT-NMR (Varian, CA, USA) and JNM-ECZ400S 400 MHz FT-NMR (JEOL Ltd., Tokyo, Japan). Chloroform-*d*/acetone-*d*_6_/methanol-*d*_4_/dimethyl sulfoxide-*d*_6_ were used as NMR solvents. Chemical shifts (δ) were reported in parts per million (ppm) downfield relative to tetramethylsilane (TMS). The following pattern was used for ^1^H NMR data: δ value (multiplicity (singlet = s), (doublet = d), (triplet = t), (quartet = q), (doublet of doublet = dd), and (multiplet = m), number of protons, coupling constants (J) quoted in Hz). Mass spectra analyses were taken on a JMS-700 (JEOL Ltd.) spectrometer at the central laboratory of Kangwon National University. Melting points were uncorrected and observed on a MEL-TEMP II (Triad Scientific, Manasquan, NJ, USA) open capillary melting points apparatus. All reactions were monitored through thin-layer chromatography utilizing Merck silica gel plates 60 F_254_ and spots visualization was accomplished by UV light absorption and/or *p*-anisaldehyde/phosphomolybdic acid stain. Silica gel 60 (230–400 mesh (40–63 μm), Merck, Darmstadt, Germany) was employed for chromatographic purification. 

Antioxidant activity was tested in the dark room by ABTS assay [30]. UV-1800 (Shimadzu Corporation, Kyoto, Japan) was used to measure UV absorption. Hellma’s 104.600-QC Quartz cell was used as a UV cell. IC_50_ values were obtained using OriginPro 8.0 software (OriginLab Corporation, Northampton, MA, USA).

Anti-inflammatory activity was measured as the effect of compounds on inhibiting NO production versus cell viability [31]. Their inhibitory effect was also tested for pro-inflammatory cytokine gene expression. 

Detailed synthetic methods and characterization of compounds and assay methods are included in Appendix A. 

## 4. Conclusions

Six new extended aromatic resveratrol analogues containing naphthalene (**2**) and its bioisosteres quinoline (**3** and **4**), isoquinoline (**5**) quinoxaline (**6**) and quinazoline (**7**) scaffolds were designed using an annulation strategy and investigated for their in vitro ABTS radical scavenging activity. All compounds exhibited enhanced antioxidant activity in the ABTS test (IC_50_ = 2.81~3.61 μM) compared with **1**. Besides, **2**–**7** also showed dose-dependent inhibition of NO production in LPS-treated RAW 264.7 cells. Analogues **5** (ED_50_ = 13.08 μM; *i*EI = 4.84) and **7** (ED_50_ = 21.67 μM; *i*EI = 4.27) furnished superior activity than **1** (ED_50_ = 26.89 μM; *i*EI = 3.02). The inhibition potency was further evidenced by the downregulation of the pro-inflammatory cytokine (IL-1β and IL-6) gene expression of activated macrophage cells. Though further research is required to explore the exact mechanisms of **5**-induced regulation of inflammatory signaling pathway and antioxidant activity, our study indicates that isoquinoline analogue **5** may show therapeutic potential in both antioxidant and anti-inflammatory therapy. Our research verified that extended aromaticity renders better antioxidant activity. In addition to extended aromaticity, nitrogen substitution gives better cell viability regardless of location and number of nitrogen atoms. Therefore, it suggests that in order to improve the biological activity of resveratrol analogues, aromaticity extension and nitrogen substitution could be a viable method for the design of future drug candidates. We are currently working on the substitution of heteroatoms other than nitrogen in the extended aromatic resveratrol.

## Data Availability

Data is contained within the article or Appendix A.

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
