# Peer review of "Design and Synthesis of π-Extended Resveratrol Analogues and In Vitro Antioxidant and Anti-Inflammatory Activity Evaluation"

_molecules, 2021, doi:10.3390/molecules26030646_

Round 1
Reviewer 1 Report
The article is very interesting and it is well readable, moreover methods and results are clearly presented.
In the introduction section the authors should also referr to the role of nitrogen aromatic ring for increasing the bioavailability of resveratrol analogues.
Regarding the antioxidant activity the authors should also perform a statiscal analysis regarding the higher antioxidant capacity of compounds 2-7 compared to compound 1.
Moreover, I advise the authors to continue their studies regarding the new compounds and to assess their antioxidant activity by means of other assays (DPPH, reducing power assay, chelating activity upon ferrous ions, ORAC methods).
Author Response
Dear Reviewer 1,
Thank you for your helpful review. I attached the author's reply to the comments of reviewer 1. Please see the attachment. I hope you would find it satisfactory to all of your queries.

Reviewer 2 Report
Rewiev of "Design and Synthesis of π-extended resveratrol analogues and in vitro antioxidant and anti-inflammatory activity evaluation" by Ki Yoon Nam et al.
The authors clearly present the background and purpose of the research. Very well prepared and presented synthetic part. Correctly planned and carried out research on the synthesis of the π-extended resveratrol analogues library and on their anti-inflammatory and antioxidant activity.
English correct.
Some comments and requests:
1) As I understand it, all chemical compounds (2-7) are new compounds? I miss this emphasis in the text, although it results indirectly from the sentence in lines 81-83.
2) Lines 61, 70: The number 1 should be bold as 1.
3) Figure 1. Please number the carbon atoms at least in resveratrol (1). This will facilitate the receipt of the article.
4) Line 108: "3-Benzyloxy acetanilide" or "3-Benzyloxy-acetanilide" or "3-Benzyloxyacetanilide"?
5) Line 111: there should be "°C".
6) A reference substance, e.g. ascorbic acid, is missing in the antioxidant tests. It is correct to compare the obtained derivatives with the starting resveratrol, but a reference substance would enrich the research. Please add.
7) It is a pity that the authors chose a single method for testing antioxidant properties. They could also use the DPPH radical method, which is also easy, cheap and fast. This could also help determine the mechanism by which free radicals are scavenged by the resulting compounds 2-7.
8) Supplementary Materials: FT-IR and UV-VIS spectra of compounds 2-7 are missing. This would complete the spectroscopic characteristics of the obtained resveratrol derivatives. Please add.
Author Response
Dear Reviewer 2,
Thank you for your helpful review. I attached the author's reply to the comments of reviewer 2. Please see the attachment. I hope you would find it satisfactory to all of your queries.
